# The evolution of scientific literature as metastable knowledge states

**Sai Dileep Koneru**[1]*, **David Rench McCauley**[2], **Michael C. Smith**[2], **David Guarrera**[2], **Jenn Robinson**[2], **Sarah Rajtmajer**[1]

**1** The Pennsylvania State University, University Park, PA, United States of America, **2** Ernst & Young, McLean, VA, United States of America

* sdk96@psu.edu

## Abstract

The problem of identifying common concepts in the sciences and deciding when new ideas have emerged is an open one. Metascience researchers have sought to formalize principles underlying stages in the life cycle of scientific research, understand how knowledge is transferred between scientists and stakeholders, and explain how new ideas are generated and take hold. Here, we model the state of scientific knowledge immediately preceding new directions of research as a metastable state and the creation of new concepts as combinatorial innovation. Through a novel approach combining natural language clustering and citation graph analysis, we predict the evolution of ideas over time and thus connect a single scientific article to past and future concepts in a way that goes beyond traditional citation and reference connections.

## Introduction

Early work in metascience can be traced back at least half a century [1], although it has been only in the last decade or so that a robust literature has been seeded exploring co-authorship networks, citation networks, topical networks and similar static and one-dimensional representations of complex interactions amongst researchers and their work. Much of this has been powered by the increased availability of digital data about scientific processes, improvements in information retrieval, network science, machine learning, and computational power. A substantial subset of this literature has focused on quantifying and predicting success in publishing—how we should measure success, who will have it, and what factors contribute to having it. Seminal work has focused on modeling citation patterns for papers [2] and researchers [3], with more recent work setting out to explain hot streaks in researchers' career trajectories [4], unique patterns of productivity and collaboration amongst the scientific elite [5], and even the role of luck in driving scientific success [6, 7]. We are also seeing the emergence of metascience as a social movement [8], catalyzed by the last decade's reproducibility crisis [9], aiming to describe and evaluate science at a macro scale in order to diagnose biases in research practice [10, 11], highlight flaws in publication processes [12], understand how researchers select new work to pursue [13, 14], identify opportunities for increased efficiency (e.g., automated hypothesis generation [15]), and forecast the emergence of research topics [16, 17].

**Data Availability Statement:** The dataset used for this work contains the titles, abstracts, references, and DOIs of the publications collected from Clarivate, but restrictions apply to the availability of these data. The data were used under special

privileges as a part of NCSES license for the current study, and so are not publicly available. The complete dataset is however available from the authors upon reasonable request and with permission of Clarivate (can be contacted by https://clarivate.com/contact-us/). However, the dataset can be fully created by using Semantic Scholar, which is free to use, using the DOIs and titles of the seed publications used for this manuscript. The DOIs and titles are available at https://github.com/QS-2/VESPID/blob/main/seed_data/score_papers.csv.

**Funding:** This research was supported by the National Center for Science and Engineering Statistics (NCSES) at the National Science Foundation through award 49100420C0030. The funders had no role in study design, data collection and analysis, decision to publish, or preparation of the manuscript.

**Competing interests:** The authors have declared that no competing interests exist.

The modern philosophy of evolution of science is rooted in Kuhn's *structure of scientific revolutions* [18]. According to Kuhn, majority of science progresses in the phase of *Normal Science* in which literature is built based on existing paradigms that constitute scientific theories, methods, techniques that are widely accepted in a research community. This phase is disrupted by shifts in paradigms caused by new scientific theories. This revolutionary characteristic of evolution of science was challenged by *parsonianism* [19]. According to this school of thought, scientific theories are composed of autonomous and distinctive parts such as concepts, topics, methods, methodological assumptions etc. Due to this, the shifts in science are piecemeal and complex rather than linear change as proposed by Kuhn. Inductive analysis of literature from interdisciplinary fields has shown that evolution in science is primarily combinatorial with splitting and merging of knowledge groups [20]. A knowledge group can split when sub-groups within them emerge due to their specialization/maturity. Merging events occur due to convergence of applied and basic sciences. This further solidifies the view among researchers in scientometric communities that science is an ever expanding combination of ideas [11, 21].

Prior work on the evolution of research can be broadly viewed in three categories based on method: network-based; language-based; and hybrid methods using both networks and language. Language-based methods include topic modeling, e.g., Latent Dirichlet Allocation (LDA) [22, 23], sequential topic modeling [17], keyword tracking [24], and analyzing linguistic context [16]. Studies using network-based methods typically use citation networks and clustering algorithms to model the literature [25, 26]. Others have used temporal [27] or multiplex [28] networks, or projections of citation networks, e.g., co-authorship graphs [29, 30]. Due to the nature of citations and citation practises, the citation analyses have various limitations. These limitations range from practical such as completeness of citations captured by the databases [31], to more fundamental issues such as accuracy of citations [32]. The other limitations include the differences in citation citation patterns among disciplines [33]. More critically, while valuable, one can not solely rely on citations for any analysis [34]. More recent work using hybrid approaches to explain bibliometric dynamics has relied on network analysis, with post-hoc application of linguistic analysis to generate explanatory labels [35]. Others have used LDA to generate topic co-occurrence networks [36]. However, to the best of our knowledge, there are no existing hybrid methods which systematically incorporate insights from both language models and citation networks for the purposes of explaining and predicting the evolution of scientific literature.

We suggest that integration of citation-based network information and semantic information using deep learning based language embeddings offer a novel opportunity to capture the trajectory of ideas within and between disciplines over time. Specifically, we show that citation-driven and language-driven models, respectively, capture overlapping but distinct and complementary dynamics in scientific research. Furthermore, we believe that this approach also avoids the reliance of pre-defined discipline categories by the scientific publishing databases. We use pre-trained neural network models [37] to generate vectorized representations of the literature while separately leveraging citation network measures (e.g., betweenness centrality), and combine these two inputs to build predictive models of topical evolution. The intuition behind the mechanisms explored herein is that scientific disciplines can be described at a high level by aggregation of related ideas. When a discipline is beginning to show signs of fracture or change via the emergence or synthesis of new ideas, we model this moment borrowing from physics the concept of *metastability*: a state easily perturbed into a new state. We suggest that integration of knowledge from different fields is a driver of this change and hence measures of *interdisciplinarity* may be indicators of metastability and thus useful predictors of change.

Recent work has highlighted the role of interdisciplinarity in scientific practice [38–41]. Interdisciplinarity has been shown to be linked to innovation and impact [11, 42]. Calls for collaboration across disciplines are prominent throughout research institutions and funding agencies (See, e.g., the U.S. National Science Foundation's Growing Convergence Research program: https://www.nsf.gov/od/oia/growing-convergence-research/index.jsp) but some have argued that the promises of interdisciplinarity are overstated and misplaced [43]. The bibliometric community has offered a data-driven framing for interdisciplinary studies, e.g., defining interdisciplinarity as a process of integrating different bodies of knowledge [44, 45].

Definitions of interdisciplinarity vary in the literature [46]. Most fall within one of two types: subject-based and network-based definitions [46]. Subject-based metrics rely on multi-classification systems to calculate interdisciplinarity, leaning on pre-defined subject categories, e.g., handed down from journals or from the Web of Science (WoS) [47]. Definitions of inter-disciplinarity are operationalized by way of these categories, e.g., percentage of references cited from outside a journals' interest categories [48, 49]. In some cases, interdisciplinary metrics are borrowed from other fields, such as the Gini index from economics or Shannon entropy from information theory [50]); these are also based on subject categories. Network-based interdisciplinarity metrics are typically assessed based on the location of a publication in a citation network [51], with centrality measures frequently being the focus. For example, between-ness centrality, which is independent of third-party categorization, was one of the first metrics used in this way [51, 52] and has likewise been used to predict future network trends [53, 54].

To study the evolution of knowledge the scientific literature, we: (1) develop methods that utilize transformers-based language models and unsupervised clustering to track the evolution of ideas over time; (2) quantify interdisciplinarity using complementary text- and citation-based metrics; and (3) explore the utility of metastability, measured through interdisciplinarity, as a predictor of scientific evolutionary events.

## Materials and methods

### Dataset

Our dataset contains detailed records of 19,177 scientific papers published in the years 2011 through 2018, with 2300 to 2500 papers for each year, representing a substantial stratified random sample of papers published in 62 prominent journals from the following social and behavioral science disciplines as strata: Criminology; Economics and Finance; Education; Health; Management; Marketing and Organizational Behavior; Political Science; Psychology; Public Administration; and Sociology. Sampling was done in conjunction with DARPA's SCORE program. The sample contains approximately equal representation of papers per year and per journal. For a complete listing of journals and sampling methodology see [55]. The semantic analysis of this work is entirely dependent on studying these papers, which will henceforth be referred to as the core publications. Metadata for these papers was collected using the Web of Science as a primary source. Digital Object Identifiers (DOIs) were used to merge WoS records with Semantic Scholar (S2) records [56, 57] for completeness of metadata coverage and author name disambiguation. When DOIs were not available from WoS, we used Crossref [58] to fill in missing DOIs for more complete record linking between WoS and S2. For citation network analyses, we also included all papers referenced by these core papers. In total, the citation network includes records of 19,177 papers and their references (819,919 papers) and about 1.45 million citations. The citations earned by the core papers were not included in the analysis since this work uses historical information to forecast the state of a knowledge group in the following year. As citations take time to accumulate, we assume that they wouldn't be available at the time of forecasting. Additionally, considering the number of

citations received would skew the findings as the publications published at the start of an year tend to garner higher citations than the ones published at the end of the year. Since it takes time for citations to accumulate, we assume that publications' incoming citations wouldn't be known available at the time of forecasting.

## Methods

We use parallel workflows to model dynamics in bibliometric data—one based on text and one based on citation networks (Fig 1). For each, we derive a measure of interdisciplinarity useful for prediction of knowledge evolution. We describe our explanatory and predictive experiments to evaluate our measures.

### SPECTER-based topic modeling

We use language embeddings-based topic modeling to identify topics within our corpus by year. To do so, we extract embeddings for each publication in our dataset using the concatenated title and abstract as an input to SPECTER (Scientific Paper Embeddings using Citation-informed TransformERs) [37], a Bidirectional Encoder Representations from Transformers [59] (BERT)-based model for generating document-level embeddings of scientific documents via pre-training on scientific papers and their citation graphs. Specifically, we use the `huggingface` implementation [60] of the pre-trained SPECTER model. SPECTER embeddings have been shown to be useful in downstream document-level tasks –citation prediction, document classification and recommendation– without any task specific fine-tuning [37].

To identify disciplines and subdisciplines, we use an unsupervised, non-parametric, hierarchical clustering algorithm, Hierarchical Density-Based Spatial Clustering of Applications with Noise (HDBSCAN) [61]. Specifically, we soft-cluster SPECTER embeddings to reflect that papers may belong to multiple (sub)disciplines with different probabilities. Because the performance of HDBSCAN generally reduces as the dimensionality of input data increases, we use UMAP [62] to reduce the dimensionality of SPECTER embeddings prior to clustering with HDBSCAN. We use multi-objective Bayesian hyperparameter tuning [63] for the UMAP-HDBSCAN pipeline to balance five evaluative criteria related to balancing inter- vs.

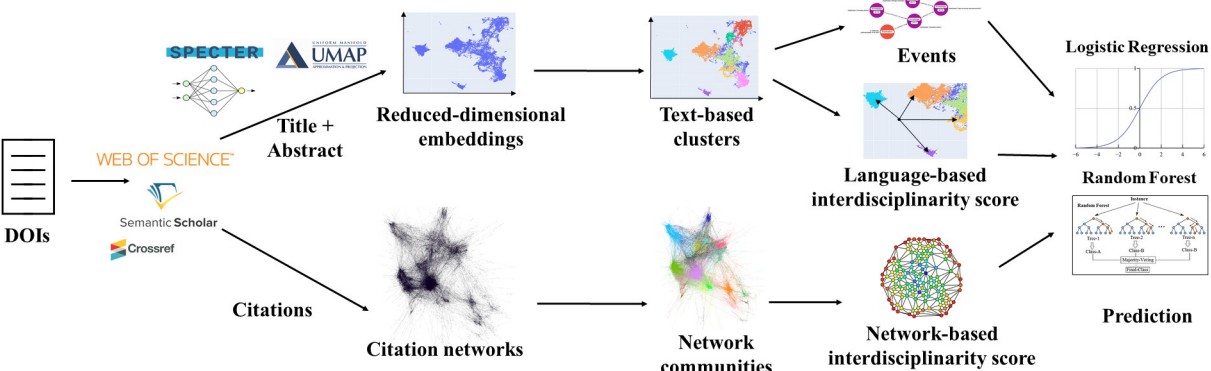

**Fig 1. Data analysis workflows.** (Top) **Text-based analysis**. Title and abstract are concatenated and input to a language embedding model, then dimensionally reduced and fed into a clustering algorithm; clusters of embedded papers are then used for event modeling and interdisciplinarity scoring. (Bottom) **Citation-based analysis**. Citation information is used to create undirected citation graphs; the Louvain algorithm is used to identify network communities and betweenness centrality is used for interdisciplinarity scoring. Interdisciplinary metrics are jointly used to predict disciplinary evolution.

intra-cluster density, number of clusters, and persistence of clusters over multiple runs of the algorithm.

In particular, UMAP requires us to specify: (1) the dimensionality of reduced embeddings; (2) a distance metric; (3) the size of the local neighborhood; and (4) the minimum separation distance for dimensionally-reduced points. For HDBSCAN, we need to provide: (5) minimum cluster size; (6) a cluster selection method; and (7) minimum number of neighboring points required to identify the "core points" of a cluster. To do so, we use the `optuna` framework for multi-objective Bayesian hyperparameter tuning [64].

For each year in the dataset, we run 1000 hyperparameter permutations for the UMAP-HDBSCAN modeling pipeline and choose hyperparameter combinations that rank highest. As UMAP is a stochastic algorithm, we perform repeated runs for each hyperparameter combination to select solutions that are robust to randomness based on the following five evaluative criteria:

- **Mean Density-Based Clustering Validation**. Density-Based Clustering Validation (DBCV) is a relative validity index that is commonly used to assess the quality of clusters obtained by density based clustering algorithms. DBCV is computed using the inter- and intra-cluster density connectedness [65]. High quality clustering solutions have high DBCV scores as they have low inter-cluster vs intra-cluster connectivity. DBCV is a commonly-used method to evaluate cluster solutions with HDBSCAN;

- **Standard deviation of DBCV**. To complement mean DBCV, we consider variation in DBCV scores over multiple runs. Robust hyperparameter combinations should exhibit low standard deviation of DBCV scores;

- **Mean number of clusters**. Evaluating cluster solutions solely on DBCV-based metrics is insufficient as solutions with few clusters or containing clusters with low persistence scores can lead to results with high mean DBCV score but negligible practical value for many domains. We therefore consider additional metrics, the first being mean number of clusters, with this formulation preferring greater cluster resolution. For consistency across metrics and to avoid the effect of different scales on the solution quality, we use min-max scaling to normalize the number of clusters when hyperparameter tuning;

- **Standard deviation of number of clusters**. As with DBCV, a good hyperparameter set should exhibit low standard deviation of cluster counts;

- **Mean of mean cluster persistence**. Optimizing for a solution with a high number of clusters can lead to ephemeral clusters that dilute results. We use the mean of mean per-cluster persistence across identical runs to evaluate stability of the clusters identified.

Given the evaluation criteria, the maximum value of the objective function occurs when the values of *Mean DBCV, Mean of mean cluster persistence, Normalized mean number of clusters* are close to 1 while the *standard deviations of DBCV, number of clusters* are close to 0. Successfully clustered papers are considered "strong members" of their cluster. While, we refer to papers that cannot be confidently assigned by the clustering algorithm as "weak members". We assign each weak member to the cluster with which it has the highest semantic similarity. For completeness, we report downstream analyses with and without inclusion of weak members. We consider this distinction because we suggest that weak members represent research which is significantly different (and potentially truly innovative) relative to existing disciplines, and as such can help explain shifts in the trajectories of fields. HDBSCAN refers to these non-confident assignments as noise; however, we expect these not to be noise in the traditional sense (e.g., an outlier or data worthy of discarding as it provides no analytical value) but

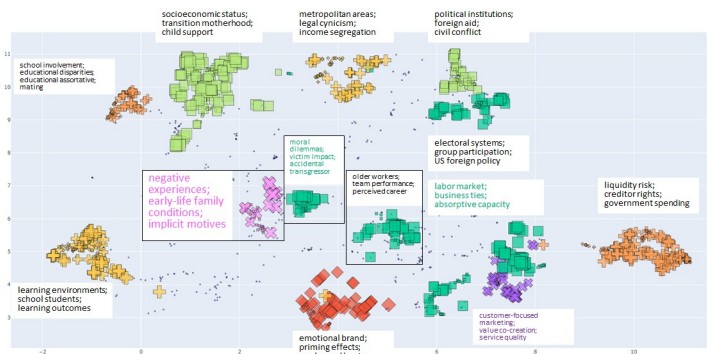

**Fig 2. UMAP projection of SPECTER embeddings of publications, projected to two dimensions, in the year 2011 colored by HDBSCAN-generated cluster labels with corresponding cluster-level keyphrases.** Each cluster plotted here contains at least 2.5% of total papers for the year and the size of each point is proportional to that publication's language-based interdisciplinarity score. Small blue points represent weak members. Note that most clusters shown are well-separated and not homogeneous in shape, suggesting that UMAP is doing a good job of dimensionally reducing the feature space in such a way that it is reasonably straightforward to partition and that a variable-density-based clustering algorithm, such as HDBSCAN, is well-suited to identifying clusters in such a dataset.

instead to potentially add value as research different from other work in their field. For example, one weak member is an article titled *Strengthening the Experimenter's Toolbox: Statistical Estimation of Internal Validity* with devoted a major portion of article to statistical methods applied to political science field is published in *American Journal of Political Science* [66] that predominantly publish practical applications.

For each cluster, we generate representative keyphrases using a procedure similar to the KeyBERT library [67], with modifications (e.g., more performant aggregation of embeddings from large numbers of documents belonging to the same cluster). Deriving keyphrases provides explanatory power for clusters and adds more nuanced understanding of the clusters than other commonly used approaches to grouping knowledge products, e.g., WoS categories. As an example, clusters identified in our dataset for the year 2011 and their corresponding keyphrases are shown in Fig 2. We identify a total of 371 clusters over the complete dataset, i.e., years 2011 through 2018. The distribution of the number of clusters for each year is shown in Fig 5.

## Citation graphs and communities

Per common practice, our citation-based analysis considers the citation network wherein nodes in the graph represent papers in our dataset and undirected edges represent citation relationships. We detect communities in this network using the Louvain community detection algorithm [68] which maximizes modularity of the network, namely the expected value of inter- vs intra-community edges [69]. Specifically, for a given time window/year of interest $t$ we consider the subgraph $G(t)$ containing only papers published in year $t$ and earlier, as well as their references. This approach allows us to make predictions for past papers without fear that future papers citing them will cause information leakage into the dataset (e.g., a model trying to predict the evolution of an idea tied to a paper from 2017 should not have access to information about papers from 2018 citing it during model training). An example of the community structure discovered via the Louvain method is shown in Fig 3.

## Quantifying interdisciplinarity

**Language-based interdisciplinarity.** Our text-based interdisciplinarity (ID) metric scores each publication based on its soft clustering membership probabilities (i.e. the probability of a

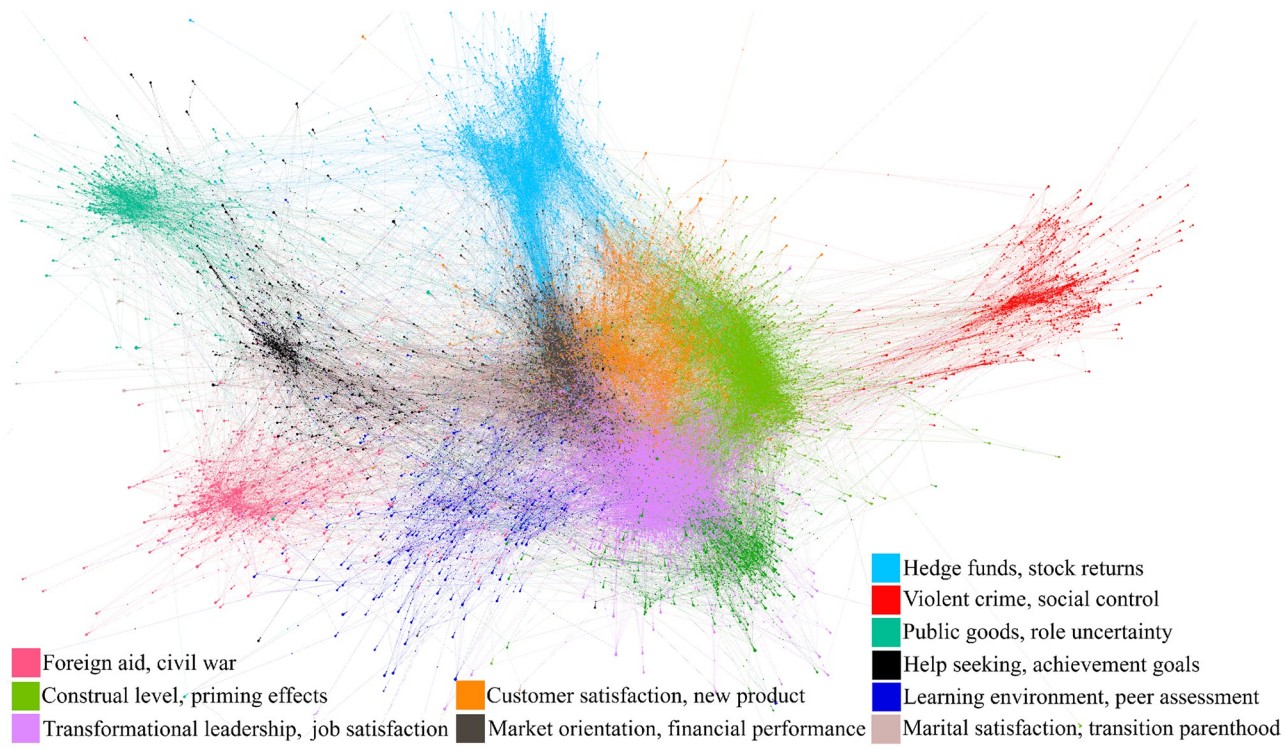

**Fig 3. An exemplary snapshot of the dense network and communities found by the Louvain community detection algorithm for the year 2011.**
Publications belonging to the communities comprising less than 2.5% of total papers for the year are suppressed to improve the visual clarity. Note that a clear community structure can be observed for this graph-only approach much like it was for the language-only clustering presented earlier.

publication belonging to each possible cluster identified by standard or "hard" clustering), considering only strong member publications. It does so by assuming that one representation of interdisciplinarity is the diversity of language pulled from different fields. This metric is calculated using Eq 1 which considers the spread in its cluster assignment probabilities. Formally:

$$\text{ID}_{text} = \frac{N}{N-1}(1 - P_{wm} - \max(P_{cluster}))\left(1 - \sigma_p\right) \tag{1}$$

where $N$ is the total number of clusters in the dataset, $P_{cluster}$ is the probability of the paper belonging to a cluster, $P_{wm}$ is the probability of the paper being a weak member of any cluster, and $\sigma_p$ the standard deviation of $P_{cluster}$ over all clusters. This formulation is more intuitive when extreme cases are considered. For example, consider a corpus with 9 clusters for the year of interest. Consider a paper that sits very clearly within a single well-defined scientific discipline, i.e., $max(P_{cluster}) = 1$ for a single cluster (consequently, $P_{wm} = 0$). The interdisciplinarity score for that paper would be $ID_{text} = 0.0$. Alternatively, imagine a paper with membership probabilities that are equivalent for all clusters, with the same probability that it may be a weak member, i.e., $P_{wm} = P_{cluster,i} = 0.1$ for $N = 9$. This would result in $ID_{text} = 0.9$, reflecting that the paper belongs to a wide array of disciplines/clusters equally, but also there is some chance that it may be a weak member—which can also be interpreted as a global uncertainty in the membership probabilities—thus keeping it from achieving a score of 1.0. This proposed interdisciplinarity metric presents a departure from existing metrics in the literature, as it allows for language clusters to potentially denote knowledge groups at different levels of granularity, ranging from topics to entire disciplines. Despite the research community's lack of consensus regarding the definition

and operationalization of interdisciplinarity, it is generally acknowledged that the concept involves the synthesis and integration of knowledge across diverse knowledge groups [46]. Thus, we expect that the proposed language-based metric can serve as an effective means of measuring interdisciplinarity, independent of pre-defined disciplinary boundaries.

**Citation-based interdisciplinarity.** We use betweenness centrality for each publication in the network as an interdisciplinarity metric, with higher centrality generally indicating higher interdisciplinarity, as has been done in previous literature [70]. As we do for community detection, we use time-windowed subgraphs for centrality measurement. Betweenness centrality is lightly modified for use as an ID metric, normalized on a [0, 1] scale. For paper $i$ in publication year $t$:

$$\text{ID}_{network} = centrality_{t,i} / max(\{centrality_t\}) \qquad (2)$$

where $\{centrality_t\}$ is the set of all centrality values for papers published in calendar year $t$.

## Text-based dynamic event modeling

We identify and track critical knowledge evolution events borrowing from the literature tracking communities in dynamic social networks [71]. Specifically, representative embeddings for each cluster are calculated using the element-wise mean of embeddings of the papers in each cluster, and clusters are compared across consecutive years by calculating the pairwise cosine similarity of the embeddings of each $[C_t, C_{t+1}]$ pair of clusters in years $t$ and $t + 1$ [71]. We then link a cluster with its best-matching cluster(s) in the consecutive time step if the cosine similarity is above 0.95 (To gain a deeper insight into the impact of the threshold value on the results, a posthoc analysis was carried out, and the findings are comprehensively presented in S1 Appendix). We employ the following taxonomy [71]:

- A *birth* event is identified at time $t$ when a cluster at time $t$ has no matching cluster(s) at time $t − 1$.

- A *death* event is identified at time $t$ when a cluster at time $t$ has no matching cluster(s) at time $t + 1$. This framework uses comparison of clusters from consecutive years to determine if a cluster is *dead* and does not consider the possibility of the revival of a dead cluster in later years. Although the reappearance of dead clusters has not been identified in this dataset, this may be explored in future work.

- Multiple clusters have *merged* at time $t$ when one cluster at time $t$ matches to two or more clusters at time $t − 1$.

- Multiple clusters have *split* at time $t$ when one or more clusters at time $t$ match to a single cluster at time $t − 1$.

- A *continuation* event is observed when one cluster at time $t$ is matched to exactly one cluster at time $t + 1$.

We group these events into two types for subsequent analyses: (1) *dynamic*—split or merge and (2) *stable*—continuity or death. Not only does treating splits and merges as a single class emerge from our metastability mental model but, given that they often co-occur, this treatment creates non-overlapping classes. We disregard birth events at present since a birth event has no preceding data from which to build a model and is unrelated to the concept of combinatorial innovation being described by metastability. Fig 4 gives a notional example of merge and continuation events. We note that events may occur in combination; e.g., a cluster may split into two, and those two clusters may simultaneously merge with two other clusters.

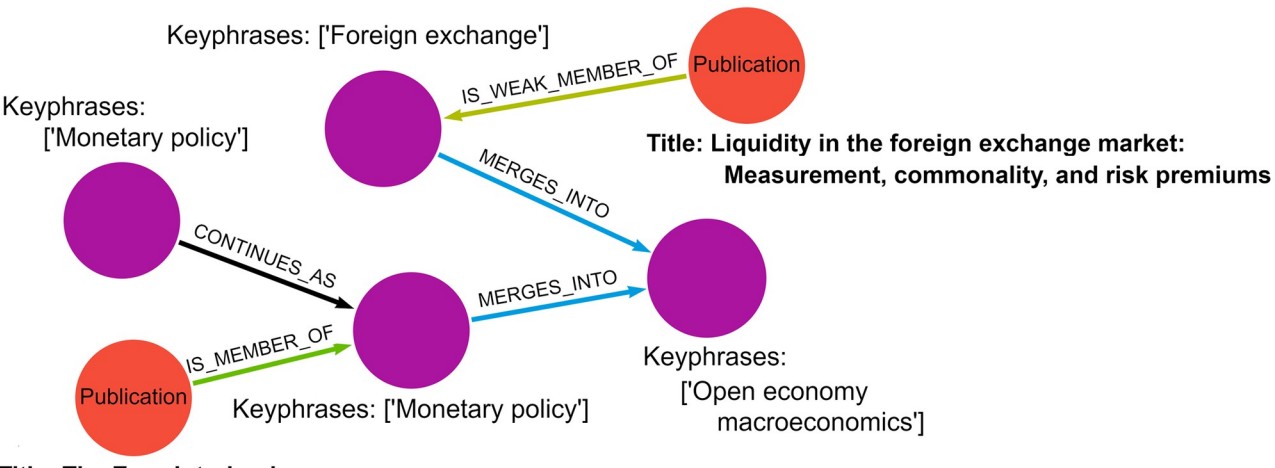

**Fig 4. Notional continuation and merge events showing weak (significantly different from existing clusters) and strong members (high confidence in its membership) of each cluster.**

## Event-tracking and prediction

We hypothesize that interdisciplinarity scores and cluster size are indicators of metastability and therefore can be used to predict cluster evolution, i.e., *dynamic* vs. *stable* events, as an endogenous and target variable. In particular, for each language cluster $C_t$ at time $t$, we use as exogenous model inputs: cluster-wise mean language-based interdisciplinarity score (which does include weak member papers); mean citation-based interdisciplinarity score for weak and strong members, treated as separate features in order to discern if there is any difference in predictive power considering weak members; and number of weak and strong member papers in the cluster.

To choose the most powerful features and test their predictive power (and thus value for further analyses), we use multivariate logistic regression and a Random Forest classifier with a binary target $\vec{y}$ representing if a dynamic event type (split or merge) is observed for a cluster at time $t + 1$ as shown in Eq 3.

$$\vec{y} = \begin{bmatrix} split/merge \\ continuation/death \end{bmatrix} \tag{3}$$

We use the entire dataset with multivariate logistic regression for explanatory power. For the random forest, we have also evaluated the model performance with different test train splits as discussed in S1 Appendix. Based on these findings, we use cluster events in 2011–17 for training and 2018 for testing, resulting in roughly an 86%/14% train/test split by cluster count with 275 events for training (split/merge: 136; continuation/death: 139) and 43 testing events (split/merge: 21, continuation/death: 22). Using the above input features and event types in year $t + 1$, we fit a random forest model using the `scikit-learn` python library [72]. To avoid overfitting, we tune hyperparameters—number of trees and maximum depth of each tree—of the random forest model using grid search and found that a model with 44 classifiers each with a maximum depth of 4 achieves the best F1 scores on held-out data. Results of these experiments are described in S1 Appendix.

## Results

Following, we show that language and network frameworks capture different information by comparing the overlap between clusters identified using text and citation-based communities. We then further investigate the nature of the information provided by both frameworks by discussing how these representations, when considered together, serve to predict the evolution of disciplines and sub-fields.

### Comparing clusters and communities suggests valuable incomplete overlap

Fig 3 gives a snapshot of network communities in 2011; comparison with Fig 2 illustrates differences in grouping across the two approaches. In general, the Louvain algorithm detects communities in the citation network at a finer resolution than our text-based clustering. For reference, Fig 5 shows the number of clusters and communities in our dataset, in addition to a measure of overlap between the two that we describe below. The number of network communities generally decreases over time, reflecting a more integrated citation graph emerging amongst the papers in our sample.

As both our language- and citation-based frameworks are unsupervised, to compare them we need to identify clusters with one another across frameworks. For this, we measure pairwise Jaccard similarity between clusters and communities, effectively looking at the fraction of shared publications between every language cluster and every network community relative to their total number of member papers. If the similarity between a cluster and a community is above 0.1 then we consider them similar. This threshold-based method (and the 0.1 threshold specifically) has been used in the literature for tracking clusters and communities over time [71, 73] and performs well across a variety of synthetic graphs. Going back to Fig 5, the inset shows the percentage of language clusters with similar (Jaccard similarity > 0.1) network

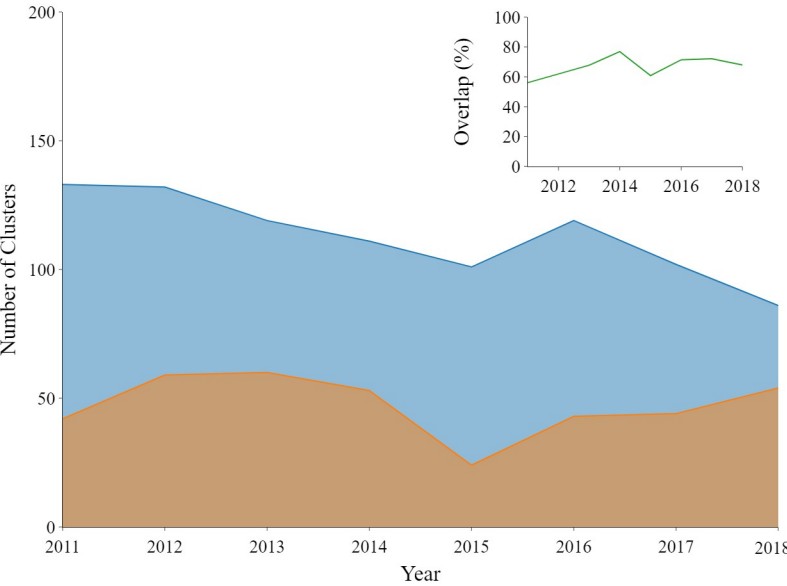

**Fig 5. Plot with number of clusters/communities identified by text-based (brown) and networks-based (blue) frameworks with inset plot showing percentage of language clusters associated with at least one network-derived community.** Note that overlap values are consistently below 100% but well above 0%, suggesting unique and complementary insights added by each. The trends in the number of language clusters, network communities by year could potentially be attributed due to the integration of knowledge from the disciplines that were not included in the dataset (refer *Discussion* section).

communities. While there is overlap between the communities and clusters, the overlap is not complete, suggesting that each approach adds unique insight.

## Illustrating knowledge evolution events

To illustrate the types of knowledge events we identify and track in this work, let us consider an example from our dataset. Fig 6 shows the evolution of a full chain of language cluster evolutionary events over the period 2011 through 2018. Every cluster in this chain has "Business and Finance" and "Economics" as the most common WoS categories among member papers. In contrast, the keyphrases generated via our language clustering approach reflect greater resolution, including phrases like "income hedging" and "intangible capital". This chain starts with a 2011 cluster that appears related to the (then recent) U.S. housing market crisis and Great Recession. There is a strong focus on work discussing corporate governance and government spending. This focus on organizational-level finance and economics mostly continues through 2017, with only a few deviations that are more focused on overall market trends. This is epitomized by the representative paper for one of the 2016 clusters, focused on European banking. Then something happens in 2018: topics appear to shift substantially from organizational/macroeconomic concepts to research focused on individual-level spending, finance, and decision-making, as can be seen both from the keyphrases representing those linguistic clusters, as well as from the representative 2018 paper focused on accounting for consumer behaviors in investing. It is interesting to note that this timing corresponds with Richard Thaler's 2017 Nobel Prize in Economics, awarded for contributions to behavioural economics.

## Knowledge evolution is significantly associated with interdisciplinarity and weak members

We use multivariate logistic regression with the mentioned endogenous and exogenous variables to evaluate how knowledge evolution may be explained through our interdisciplinarity scores, cluster size and network metrics. Per common practice, we insert a constant and a year variable to account for potential temporal effects. We attempt to explain whether or not

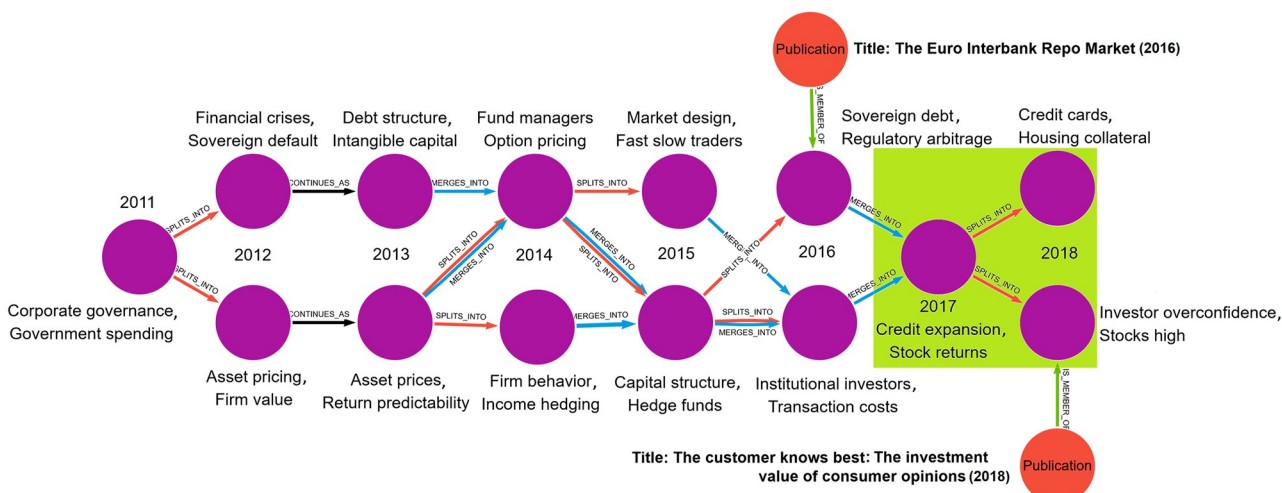

**Fig 6.** Evolution of a set of language clusters from 2011 to 2018 (left to right) and keyphrases for each, along with two representative papers for two of the clusters. Note the marked change in focus between 2016 and 2018 evidenced by representative titles and cluster keyphrases. The split event for the 2017 cluster was successfully predicted by the random forest classifier described later (brown box).

**Table 1. Multivariate logistic regression results describing associations with split or merge (1) vs. continuation or death (0).** Note significant positive associations with language score, network score, number of weak members, and a negative association for the year. All other features were not significant, and left out via purposeful selection for a more parsimonious model; see S1 Appendix.

| Model Input (per cluster) | Model estimates | | Marginal effects | |
|---|---|---|---|---|
| | Coefficient | P | Effect | P |
| Mean language ID score (strong members only) | 0.534 | 0.000 | 0.116 | 0.000 |
| Number of weak members | 0.449 | 0.003 | 0.097 | 0.002 |
| Mean network ID score (strong members only) | 0.292 | 0.030 | 0.063 | 0.025 |
| Publication year | -0.372 | 0.007 | -0.081 | 0.005 |
| Constant | -0.009 | 0.941 | | |

clusters split or merge first, in order to evaluate the strength of associations between our hypothesized inputs and outputs.

Per Table 1, we see significant positive associations between a cluster splitting or merging and the language interdisciplinarity score and network interdisciplinarity score with only certain associations (i.e., without weak members). Following common best practice, tests were conducted with all features, and, finding some insignificant, repeated with only significant features. See S1 Appendix for details of this purposeful selection. We also see positive association with the number of weak members, and a negative association with the year. Year was included per common practice to remove potential effects from time passing. Note that this model had a higher pseudo $R^2$ than a model without the year included. Future work should investigate temporal associations through, e.g., time series analyses. Though all marginal effects are on the same order of magnitude, ranking by those effects, the language interdisciplinarity score is most important, followed by the number of weak members, and the network score without weak members. Next, we further investigate this statistical relationship by testing the predictive power of a model trained on only a subset of cluster data.

## Validating our statistical result with predictive power—equal importance of interdisciplinarity scores

We have shown significant associations between knowledge splitting and merging, and interdisciplinarity and weak members. Here we go further by performing predictive modeling with a random forest classifier. Including only features shown to be statistically significant, we achieve a micro-averaged $F_1 = 0.814$ on our held-out test set, with $F_1 = 0.818$ on our class representing knowledge evolution (i.e. splitting or merging), a performance that is significantly better than random chance. Specifically, we present Table 2, which intuitively shows both interdisciplinarity scores to be equally important in achieving our predictive power. The number of weak members associated with a given cluster is next-most important, followed by the year variable. We validated against potential issues that can affect the Gini feature importance values from a random forest, specifically issues that arise when features exhibit multicollinearity and a bias towards numeric and high-cardinality categorical features [74]. The first is not a problem in this case, as the high-correlation features were removed as a result of the logistic regression analysis discussed earlier. The second is expected to only be a minimal concern for this analysis, as the only non-numeric feature in this model is the publication year. Because this is a low-cardinality categorical feature, it may be the victim of a bias in the feature importances and, as a result, the year's true ranking in the feature importance table could be higher than is indicated. As this is not a critical change in the data for our analysis, correcting for this bias is beyond the scope of this work. Taken together, our results underscore the importance of including both the linguistic and network viewpoints of interdisciplinarity. To further

**Table 2. Random forest results on a held-out test set predicting the different types of cluster events a given cluster would experience in the next year, with the same features as in Table 1.** We achieve a micro-averaged $F_1 = 0.814$ on our held-out test set, with a class-specific $F_1 = 0.818$ for the class representing knowledge evolution (splits and merges). Per reported Gini feature importance of each independent variable, both interdisciplinarity scores are equally important, followed by number of weak members, then year. Note that the sort order of this table is identical to that of Table 1 to allow for more direct comparison of logistic regression coefficients to random forest feature importances.

| Model input feature | Gini Importance |
|---|---|
| Mean language ID score (strong members only) | 0.374 |
| Number of weak members | 0.214 |
| Mean network ID score (strong members only) | 0.269 |
| Publication year | 0.143 |

validate this claim we have conducted feature ablation studies which show that classifier trained using only linguistic features or network features have lower predictive power. Details of ablation studies are provided in S1 Appendix.

## Discussion

Through both explanatory and predictive efforts, we show that language and network interdisciplinarity increase metastability of disciplines and sub-fields. Interestingly, network interdisciplinarity of strong member papers is significantly predictive of these mixing events, although the number of strong members is not. By contrast, although weak members' network interdisciplinarity is not significantly predictive, more weak members are predictive of knowledge recombination. One explanation may be that papers that do not cluster neatly are indicative of combinatorial innovation that is expressed as knowledge mixing events in our framework. Consequently, if one is interested in spurring broad interdisciplinarity, one might encourage more weakly-clustered research, regardless of its own network-derived interdisciplinarity. Future work should further investigate these relationships, in particular over longer time scales. For example, one might explore whether weak members at time $t_1$ can lead to a stable cluster at $t_2$ ($t_2 > t_1$), i.e., indicating research efforts have reached a "critical mass".

Our work motivates new hybrid models that align multiple views of the literature, e.g., linguistic, bibliometric, into unified modeling frameworks. Looking beyond traditional single-view approaches, such frameworks would be better suited to capture the richness of the scholarly record. This can be achieved through so-called graph machine learning models, which support an integrated representation of data reflecting both its content, e.g., language in the case of a scientific paper, and its context within a network. Further, the work we describe here is mostly based on unsupervised learning. There is no readily-available ground truth that is universally acknowledged to reflect the changing nature of scientific thought, disciplines, and sub-disciplines at a time scale reflective of how ideas mature and evolve. Future work should build benchmark datasets with which the metascience community can engage to evaluate and test these approaches more thoroughly than is currently possible. The building of such benchmarks is bound to be a challenge, given that organization and taxonomization of scientific knowledge can be considered along many dimensions and carries inherent subjectivity. This was evident from expert feedback we elicited to evaluate our algorithm-generated clusters. Further detail about the brief survey we conducted is provided in S1 Appendix.

Finally, there exists a number of interdisciplinary metrics in the literature and a lack of consensus about their utility [46]. Future work may consider a comparison of various interdisciplinary metrics existing in the literature, including their efficacy for downstream tasks such as understanding and predicting knowledge evolution as we do here. Likewise, there are many

clustering algorithms that may be engaged and compared for this task. These too should be compared in context. Here again, a critical challenge is lack of benchmark datasets for evaluation.

## Limitations

There exist limitations of this work inherent to both our dataset and modeling approach. The current dataset does not encompass fields of study that are closely related to yet distinct from the disciplines chosen. Consequently, publications that integrate the concepts from both these related but distinct fields may be labelled as weak members by this study, even if they belong to a knowledge cluster when publications from the closely related field are incorporated. Another limitation of the study is our use of one year time steps for identifying semantic clusters. Accordingly, we do not capture dynamics with evolutionary cycles less than one year. Additionally, we have used the purposeful selection method, that relies on statistical significance, to choose the variables which limits the scope of this study [75]. Future work aimed at providing a complete picture of the phenomena presented in this paper should evaluate relative contributions of the complete set of variables considered.

We highlight that the approach we present here is intentionally broad and conceptual. However, significant improvements in predictive performance may not be possible with a one-size-fits-all modelling approach across the scientific corpus. Rather, birth and death in one field may be catalyzed by meaningfully different factors in one field vs. another. Accordingly, future work may dig into field-specific modeling in cooperation with domain experts.

## Conclusion

This paper proposes a hybrid language- and network-based framework that uses semantic embeddings and citation information to model metastability of ideas in order to identify dynamic events associated with the rise, fall, combination, and dispersion of topics in the scholarly corpus. We show that this hybrid approach is distinct from approaches based on linguistic or citation information alone. The methods we propose rely on multiple views of interdisciplinarity as predictors of scientific knowledge transitions. Our work lays groundwork for novel approaches that bring together linguistic and citation modeling for understanding dynamics in scientific literature.

## Supporting information

**S1 Appendix.**
(PDF)

**S1 Table. Initial multivariate logistic regression results.** Results describing associations with knowledge evolution, binarized as split or merge (1) and continuation or death (0), and all our exogenous variables. Note significant positive associations with language score and number of weak members, plus a negative association for the year. Mean network interdisciplinarity score (considering strong members only) is statistically insignificant, as is the number of weak members. Mean network interdisciplinarity score (considering weak members only) is insignificant, yet it has a significant marginal effect. Per common practice, we purposefully re-ran our analysis discarding insignificant variables, to evaluate significance of network score among weak members and confirm our findings on language score, number of weak members, and year.
(PDF)

**S2 Table. Summary of results from a brief survey of three domain experts.**
(PDF)

**S1 Fig. Impact of test-train split size on F1 Score.** Plot showing the best F1 scores achieved on the test data when random forest model trained by varying the sizes of training data on the events occurring in the following year. The labels shown next to each data point represent the model parameters (maximum depth, number of classifiers). When trained on events occurring between 2011 and 2014, every model in the grid search overfit to the training data.
(TIF)

**S2 Fig. Parameter grid search: F1 score variation.** Line plot showing the random forest model F1 scores achieved when events in 2011-17 were used for training and tested on predicting 2018 events for a sample of model parameters (maximum depth, number of classifiers) from grid search. The plot shows that the maximum depth of classifier trees has a higher effect on generalizability of the model than number of classifiers. The model performance on held out data diverges from the performance on training data due to the overfitting as maximum depth increases.
(TIF)

**S3 Fig. Figure showing the impact of similarity threshold value on fraction of different event groups identified.**
(TIF)

**S4 Fig. Figure showing the impact of similarity threshold value on prediction task performance.**
(TIF)

## Acknowledgments

The authors would like to thank Dr. Ashley Arigoni for her work on cluster comparison visualizations, as well as Mr. Joe Gorney and Mr. Alex Wade of Semantic Scholar for their aid in troubleshooting data engineering issues, and Dr. Ilya Rahkovsky of the Center for Security and Emerging Technology at Georgetown University and Dr. Phoebe Wong of Quantitative Scientific Solutions for their insights on the final analyses and drafts of this paper. Additionally, we thank the anonymous reviewers for their valuable feedback which strengthened this work.

## Author Contributions

**Conceptualization:** David Guarrera, Sarah Rajtmajer.

**Data curation:** Sai Dileep Koneru, David Rench McCauley, Michael C. Smith, Sarah Rajtmajer.

**Formal analysis:** Sai Dileep Koneru, David Rench McCauley, Michael C. Smith, David Guarrera, Sarah Rajtmajer.

**Funding acquisition:** David Guarrera, Sarah Rajtmajer.

**Investigation:** Sai Dileep Koneru, David Rench McCauley, Michael C. Smith, Jenn Robinson, Sarah Rajtmajer.

**Methodology:** Sai Dileep Koneru, David Rench McCauley, Michael C. Smith, Sarah Rajtmajer.

**Project administration:** David Rench McCauley, David Guarrera, Sarah Rajtmajer.

**Resources:** David Guarrera.

**Supervision:** David Guarrera, Sarah Rajtmajer.

**Validation:** Sai Dileep Koneru, David Rench McCauley.

**Visualization:** Sai Dileep Koneru, David Rench McCauley, Michael C. Smith, Jenn Robinson.

**Writing – original draft:** Sai Dileep Koneru, David Rench McCauley, Michael C. Smith, Sarah Rajtmajer.

**Writing – review & editing:** Sai Dileep Koneru, David Rench McCauley, Michael C. Smith, David Guarrera, Jenn Robinson, Sarah Rajtmajer.

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
