## [Decision Letter · Decision Letter 0]

23 Aug 2022

PONE-D-22-12774The evolution of scientific literature as metastable knowledge statesPLOS ONE

Dear Dr. Koneru,

Thank you for submitting your manuscript to PLOS ONE. After careful consideration, we feel that it has merit but does not fully meet PLOS ONE’s publication criteria as it currently stands. Therefore, we invite you to submit a revised version of the manuscript that addresses the points raised during the review process.

We look forward to receiving your revised manuscript.

Kind regards,

Ilya Safro, Ph.D.

Academic Editor

PLOS ONE

Journal Requirements:

This research was supported by the National Center for Science and Engineering Statistics (NCSES) at the National Science Foundation through award 49100420C0030. The study funders consulted with the authors throughout the project timeline.

This research was supported by the National Center for Science and Engineering 374

Statistics (NCSES) at the National Science Foundation through award 49100420C0030.

However, funding information should not appear in the Acknowledgments section or other areas of your manuscript. We will only publish funding information present in the Funding Statement section of the online submission form. 

This research was supported by the National Center for Science and Engineering Statistics (NCSES) at the National Science Foundation through award 49100420C0030. The study funders consulted with the authors throughout the project timeline.

Reviewers' comments:

Reviewer's Responses to Questions

**Comments to the Author**

1. Is the manuscript technically sound, and do the data support the conclusions?

Reviewer #1: Partly

Reviewer #2: Yes

2. Has the statistical analysis been performed appropriately and rigorously? 

Reviewer #1: Yes

Reviewer #2: Yes

3. Have the authors made all data underlying the findings in their manuscript fully available?

Reviewer #1: No

Reviewer #2: Yes

4. Is the manuscript presented in an intelligible fashion and written in standard English?

Reviewer #1: No

Reviewer #2: Yes

5. Review Comments to the Author

Reviewer #1: To my understanding, this is a study paper. Some of the methods presented in this paper can be questionable but the study and direction of the study is interesting. There are several missing citations and grammatical errors. Hence I do advise the authors to revise the paper and correct the mistakes. The paper itself is not so strong , however the study might get researchers to think about some of the mentioned methodology which can lead to future discoveries.

Based on the taxonomy, if a death occurs at time t but reappears at time t+1, how would you explain it?

Reviewer #2: The manuscript “The evolution of scientific literature as metastable knowledge states” proposes a novel method of combining both citation- and language-based techniques to follow the changes in scientific literature. Authors use a combination of time-tested and modern methods to perform feature engineering and clustering and formulate a machine learning problem of predicting the state of a cluster in the future. They show that both language and network-derived features have significant importance in this prediction process.

Pros:

- Rigorous statistical analysis of the results

- Good pipeline summary and overall narrative

- Clear motivation

- Popular (and well-documented) frameworks, ideas and techniques used

Details I would pay additional attention to:

- Figure 3 is quite hard to read. Not clear what layout was used.

- When the authors describe their ML problem, they split the data into train/test set a single time: events from 2011-2017 are used for training and from 2018 for test. It could be that this particular split is not stable and may show different results if another option is taken (say, 2011-2015 for training and 2016 for test or something similar).

- Authors say that they used a scikit-learn Random Forest implementation with default hyperparameter values. From my experience, the provided default values tend to make RF overfit and require optimization.

- The dataset description part does not include any information about how many samples were taken from each of the "prominent journals" and how this number evolves over time. It may affect clustering results, especially in a dynamic scenario.

- Authors mention that they used “multi-objective Bayesian hyperparameter tuning for the UMAP-HDBSCAN pipeline”, but they did not include the optimization target, so it’s not quite clear what would be considered optimal. It significantly affects the clustering itself as well as its input data (from UMAP dimensionality reduction framework).

- I would like to see the number of clusters over time. The authors only provided the total number of clusters (371).

- I would also like to see ablation studies (what if only network-based or language based ID scores are used?) and corresponding classification scores.

Verdict:

It is a well-written paper with promising ideas, reasonably easy-to-follow technical part and good illustrations (for the most part). However, I believe that it is possible to get more fruitful results from the obtained preprocessed data and calculated features. I am also curious to see how this approach performs on a larger scale dataset and how the topic granularity is taken into account. Some technical details were (deliberately?) omitted by authors, but I believe they can include them if they find it necessary.

6. PLOS authors have the option to publish the peer review history of their article (what does this mean?). If published, this will include your full peer review and any attached files.

Reviewer #1: No

Reviewer #2: No

---

## [Author Response · Author response to Decision Letter 0]

17 Oct 2022

We thank the editors and the reviewers for their consideration and their thoughtful and helpful comments. We have implemented a number of revisions to the work and the paper based on your suggestions, and the paper is stronger now because of it. Following, please find point-by-point response to each of the editor's and reviewers' comments. We have also provided the response in the Response to Reviewers file.

Response to Editor's comments

Response: Thank you. The manuscript now meets the style requirements, per the link you have provided.

This research was supported by the National Center for Science and Engineering Statistics (NCSES) at the National Science Foundation through award 49100420C0030. The study funders consulted with the authors throughout the project timeline.

Response: The funders had no role in study design, data collection and analysis, decision to publish, or preparation of the manuscript. We have provided the amended Financial Disclosure Statement in our cover letter, per request.

This research was supported by the National Center for Science and Engineering Statistics (NCSES) at the National Science Foundation through award 49100420C0030.

However, funding information should not appear in the Acknowledgments section or other areas of your manuscript. We will only publish funding information present in the Funding Statement section of the online submission form. 

This research was supported by the National Center for Science and Engineering Statistics (NCSES) at the National Science Foundation through award 49100420C0030. The study funders consulted with the authors throughout the project timeline. Please include your amended statements within your cover letter; we will change the online submission form on your behalf.

Response: We have removed funding information from the Acknowledgements section of the manuscript. We have provided the amended Financial Disclosure Statement in our cover letter, per request.

b) If there are no restrictions, please upload the minimal anonymized data set necessary to replicate your study findings as either Supporting Information files or to a stable, public repository and provide us with the relevant URLs, DOIs, or accession numbers. For a list of acceptable repositories, please see %http://journals.plos.org/plosone/s/data-availability#loc-recommended-repositories.

Response: The publication DOIs used as the seeds to collect data represent a sufficient dataset to replicate our findings, either directly using Web of Science (for anyone with a license) or indirectly using Semantic Scholar (which is open source). We have added the link to the Data Availability statement in the manuscript.

Response: Thank you. We have carefully reviewed the reference list. It is complete and correct.

Response to Reviewers' Evaluations

1. Is the manuscript technically sound, and do the data support the conclusions?

Reviewer #1: Partly

Reviewer #2: Yes

Response: To further ensure the technical soundness of the work, we have added ablation studies, modified the sizes of train and test datasets, and used grid search to avoid overfitting of the classification model. These changes were based on reviewers' suggestions. Further detail on these changes is provided below, in Response to Reviewers' Comments to the Authors.

2. Has the statistical analysis been performed appropriately and rigorously? 

Reviewer #1: Yes

Reviewer #2: Yes

3. Have the authors made all data underlying the findings in their manuscript fully available?

Reviewer #1: No

Reviewer #2: Yes

Response: We have now added the data file consisting of seed DOIs that are used for collecting the data from the scientific data sources (Web of Science/Semantic Scholar). We have added the file to a public github repo and added the link to the Data Availability Statement.

4. Is the manuscript presented in an intelligible fashion and written in standard English?

Reviewer #1: No

Reviewer #2: Yes

Response: We have done a thorough proofread and copy-edit of the manuscript.

Response to Reviewers' Comments to the Authors

Reviewer #1:

1. To my understanding, this is a study paper. Some of the methods presented in this paper can be questionable but the study and direction of the study is interesting. 

Response: Thank you for the comments. We have made some changes to the methods - specifically, we have: 

1) added ablation studies -- manuscript Section S1 Appendix, and referenced in Section Results;

2) demonstrated robustness of the methods with respect to varied selection of train and test datasets -- manuscript Section S1 Appendix, and referenced in Section Materials and Methods;

3) and, used grid search to avoid overfitting of the classification model -- manuscript Section S1 Appendix, and referenced in Section Materials and Methods.

2. There are several missing citations and grammatical errors. Hence I do advise the authors to revise the paper and correct the mistakes. 

Response: We have done a thorough proofread and copy-edit of the manuscript.

3. The paper itself is not so strong, however the study might get researchers to think about some of the mentioned methodology which can lead to future discoveries.

Response: We do believe that the methodological changes and additions we have made have made the paper stronger. We also appreciate and agree with your comment on methodology. Given the bibliometrics literature's longstanding focus on citation networks, we highlight the importance of novel methods for modeling text and networks together for understanding scientific trends and dynamics.

4. Based on the taxonomy, if a death occurs at time t but reappears at time t+1, how would you explain it?

Response: Thank you for the interesting question. You are right -- it is possible that a knowledge cluster (topic/discipline) could reappear at a future timestep after it has ''died''. We did not investigate this possibility when designing the current study. However, the results from the study allowed us to post-hoc manually check for the reappearance of dead clusters. To do so, we calculated pairwise cosine similarities between the representative language embeddings of all dead clusters to all clusters appearing in the later time steps. Based on the definition of the evolutionary events, we have used the similarity threshold of 0.95 to consider a dead cluster as having ''reappeared''. We did not observe any of these cases in our dataset when we did this check. However, this is a possibility that should be noted, and we have now included reference to it as a footnote in Section Materials and methods.

Reviewer #2: 

The manuscript “The evolution of scientific literature as metastable knowledge states” proposes a novel method of combining both citation- and language-based techniques to follow the changes in scientific literature. Authors use a combination of time-tested and modern methods to perform feature engineering and clustering and formulate a machine learning problem of predicting the state of a cluster in the future. They show that both language and network-derived features have significant importance in this prediction process.

Pros:

- Rigorous statistical analysis of the results

- Good pipeline summary and overall narrative

- Clear motivation

- Popular (and well-documented) frameworks, ideas and techniques used

 Details I would pay additional attention to:

1. Figure 3 is quite hard to read. Not clear what layout was used. 

Response: We believe the difficulty may have been related to all of the grey nodes in the previous image. We have revised the image suppressing the grey nodes and we have made note of this in Section Materials and methods.

2. When the authors describe their ML problem, they split the data into train/test set a single time: events from 2011-2017 are used for training and from 2018 for test. It could be that this particular split is not stable and may show different results if another option is taken (say, 2011-2015 for training and 2016 for test or something similar).

Response: We appreciate the concern. We have now conducted additional experiments to address this issue. The results of which are presented in Appendix. While training with events from 2011 to 2017 does offer best performance, training on events 2011 to 2015 also offer good performance. However, the other splits (2011 - 2014 and 2011 - 2016) lead to overfitting of models.

3. Authors say that they used a scikit-learn Random Forest implementation with default hyperparameter values. From my experience, the provided default values tend to make RF overfit and require optimization. 

Response: Thank you very much for this helpful suggestion. We have now implemented grid search to avoid overfitting and found the optimal parameters for the model. The observations from the parameter search experiments were reported in the Appendix section of the article. Using the optimal parameters has improved the prediction metrics. Thank you once again.

4. The dataset description part does not include any information about how many samples were taken from each of the ``prominent journals'' and how this number evolves over time. It may affect clustering results, especially in a dynamic scenario. 

Response: We apologize for the lack of detail here. We borrow from the sample collected in conjunction with the DARPA SCORE program. The sample contains approximately equal representation of papers per year and per journal. The complete list of journals is provided in Supplement Table 1 here: https://osf.io/preprints/socarxiv/46mnb/. This is now stated and referenced within the Dataset section of our paper.

5. Authors mention that they used ``multi-objective Bayesian hyperparameter tuning for the UMAP-HDBSCAN pipeline'' but they did not include the optimization target, so it’s not quite clear what would be considered optimal. It significantly affects the clustering itself as well as its input data (from UMAP dimensionality reduction framework).

Response: We now provide additional details related to the hyperparameter tuning process in the SPECTER-based topic modeling subsection.

6. I would like to see the number of clusters over time. The authors only provided the total number of clusters (371). 

Response: The distribution of number of language-based clusters as well as network communities is shown in the inset plot in Figure 5 of the manuscript.

7. I would also like to see ablation studies (what if only network-based or language based ID scores are used?) and corresponding classification scores.

Response: Thank you for the suggestion. We have now conducted ablation studies and reported the results in the S1 Appendix of the manuscript, and referenced them in Section Results of the main text. The results of the ablation studies show that performance of the models trained using either language- or network-based features is lower than combined performance, but also offers insights into the power of SPECTER-based language modelling.

Verdict: It is a well-written paper with promising ideas, reasonably easy-to-follow technical part and good illustrations (for the most part). However, I believe that it is possible to get more fruitful results from the obtained preprocessed data and calculated features. 

Response: We appreciate this comment and we agree that the additions you have suggested (train/test splits, ablation studies, hyperparameter tuning) have made the paper stronger now.

8. I am also curious to see how this approach performs on a larger scale dataset and how the topic granularity is taken into account.

Response: This is an interesting question. We agree that this approach should continue to be tested on other datasets and refined in future work. With respect to which datasets should be next, we have a few thoughts:

Generality: This dataset is relatively large compared to many in the bibliometrics literature. It is, therefore, also quite general to the social and behavioral sciences broadly. We would suggest that the most important next step would be to test this approach on a smaller dataset but one where some better measures of ground truth are available. In the early days of developing these methods, we tested on much smaller samples in domains where we (authors) personally had expertise, e.g., network science, machine learning. We did this so we could have a sanity check when we looked at our clusters and interdisciplinarity metrics anecdotally. We imagine that ``one-size-fits-all'' models are helpful from an intuition perspective, but are going to necessarily lack specificity.

Computational complexity: This approach should scale to significantly larger corpuses and we have indeed run the pipeline on the full Web of Science corpus starting in 2005, but at that scale (millions of papers per year) the clustering algorithm requires careful selection of hyperparameter distributions to avoid extremely long per-trial convergence times.

We have added an abbreviated version of these thoughts to the paper's Discussion section.}

9. Some technical details were (deliberately?) omitted by authors, but I believe they can include them if they find it necessary.

Response: Thank you for this suggestion. We have now added technical details about clustering and hyperparameter tuning in Section Materials and Methods. We believe that the current level of detail is sufficient to allow other researchers to replicate our approach.

---

## [Decision Letter · Decision Letter 1]

27 Dec 2022

PONE-D-22-12774R1The evolution of scientific literature as metastable knowledge statesPLOS ONE

Dear Dr. Koneru,

Thank you for submitting your manuscript to PLOS ONE. After careful consideration, we feel that it has merit but does not fully meet PLOS ONE’s publication criteria as it currently stands. Therefore, we invite you to submit a revised version of the manuscript that addresses the points raised during the review process.

We look forward to receiving your revised manuscript.

Kind regards,

Ilya Safro, Ph.D.

Academic Editor

PLOS ONE

Reviewers' comments:

Reviewer's Responses to Questions

**Comments to the Author**

1. If the authors have adequately addressed your comments raised in a previous round of review and you feel that this manuscript is now acceptable for publication, you may indicate that here to bypass the “Comments to the Author” section, enter your conflict of interest statement in the “Confidential to Editor” section, and submit your "Accept" recommendation.

Reviewer #2: All comments have been addressed

Reviewer #3: (No Response)

Reviewer #4: (No Response)

2. Is the manuscript technically sound, and do the data support the conclusions?

Reviewer #2: Yes

Reviewer #3: Yes

Reviewer #4: Yes

3. Has the statistical analysis been performed appropriately and rigorously? 

Reviewer #2: Yes

Reviewer #3: Yes

Reviewer #4: Yes

4. Have the authors made all data underlying the findings in their manuscript fully available?

Reviewer #2: Yes

Reviewer #3: No

Reviewer #4: No

5. Is the manuscript presented in an intelligible fashion and written in standard English?

Reviewer #2: Yes

Reviewer #3: Yes

Reviewer #4: Yes

6. Review Comments to the Author

Reviewer #2: This is my short review of the first revision of the manuscript “The evolution of scientific literature as metastable knowledge states”. In my opinion, the authors addressed the comments I left in my first review. From the technical standpoint, paper looks more explicit and transparent now.

However, I still have some additions considering the manuscript:

1. When the citation network is constructed, the authors use a larger dataset of papers because they aim to capture the information about references. But they only use the reference section of the papers (from what I understand from this sentence: “For citation network analyses, we also included all papers referenced by these papers.”). My concern is that this approach may not be comprehensive because it does not contain the information about the incoming connections (WHO cites a given paper, e.g. “Cited by” button in Google Scholar). I understand that this data may be harder to obtain and process, but that this improvement could be beneficial for citation network analysis.

2. I would like to get a better understanding of “weak members” of the clusters. Since the authors mention that “HDBSCAN refers to these non-confident assignments as noise”, which is true, they also have expectations about novelty these outliers can bring: “We expect these not to be noise in the traditional sense (e.g., an outlier or data worthy of discarding as it provides no analytical value) but instead to potentially add value as novel research.” I would like the authors to emphasize on this claim and maybe provide some examples of their point of view and why they think it is valid. I personally would like to see examples, where these “weak members” at timestamp t form a stable cluster at timestamp t+1. It is also interesting to investigate their stability: how they behave over time and over different clustering (UMAP + HDBSCAN) runs. These papers, being outliers, can simply be a result of some edge cases for pre-trained embeddings. For example, some research areas could be covered worse than others resulting in underrepresented “spots” in the latent space.

3. One of my previous comments was about large scale experimentation and scaling. By saying that, I mean the size of textual embedding space as well. At this moment, textual embeddings (then dimensionlality reduction and clustering) are applied to only less than 20,000 papers, which I cannot refer to as a “large scale” experiment (I am expecting at least a million papers there). The citation network scales differently, but it is still a relatively small (and very sparse) graph. It would be interesting if the authors could replicate their results using some faster clustering algorithm (i.e., k-means) and experiment with larger datasets in their future work.

Verdict: I think that he authors made some improvements with their first revision, but there are still several things to address before the manuscript can be fully accepted. If they believe that more experimentation is not feasible/realistic/necessary to perform, I would gladly ask them to add some comments on why the particular decisions were made.

Reviewer #3: This paper uses a large data set of papers and citations to examine how scientific ideas evolve over time. The paper was well explained with interesting figures. The overall results made sense and were interesting.

There was no mention of the common errors in citations in the limitations. Likely this measurement error would not be a big issue given the sample size, but it should be mentioned. See for example, DOI: 10.1042/CS20201573.

The reduction in clusters over time (figure 5) could be because of movements outside the selected fields that are not captured. For example, for the not included field “Computer Science” there could have been a cluster building in the early years that was then joined by researchers from one of the selected fields. I don’t believe that this kind of movement would be captured by this study design. It is counter-intuitive that ideas have decreased over time, and no explanation is given.

There are a lot of analytical choices in the paper, hence there is likely to be some model uncertainty (DOI: 10.1111/j.1467-9574.2012.00530.x). Other researchers may have made different and equally plausible decisions that could give different results. I think this should be acknowledged as a limitation.

This is not multinomial logistic regression (line 248) as that is for outcomes with multiple classes. The outcome here is binary, so this is simply logistic regression.

The authors are not compliant with the PLOS ONE data sharing policy: “The PLOS Data policy requires authors to make all data underlying the findings described in their manuscript fully available without restriction, with rare exception”

Minor comments

• Line 36 “state-of-the-art” is hyperbole. Point out the benefits of the methods.

• Line 80, why were these fields chosen?

• What is embedding? Line 98.

• SPECTER may “outperform” other methods, but how accurate is it? Line 101

• There are a lot of acronyms in the paper. I am fairly certain that DBCV is not defined.

• The overall proportion of weak members would be interesting (figure 2 legend)

• Figure 2 was interesting, but I didn’t know what the x- and y-axes were.

• Could some joins between fields be due to commonly used methods papers?

• I found it surprising that the death of an area was labelled as “stable”, as that seems like a definite change. This is just a wording issue.

• Figure 4. The paper for the top-right circle is missing a title.

• The splits are based on annual data. That is probably suitable, but should be mentioned as a limitation.

• “features were not significant” (table 1 legend) best to focus on the size of the difference rather than the statistical significance (DOI 10.1080/00031305.2016.1154108)

• “including only features shown to be statistically significant” there are better model selection methods available than using statistical significance, see for example Statistical Learning with Sparsity: the Lasso and Generalizations by Trevor Hastie, Robert Tibshirani and Martin Wainwright (May 2015)

Reviewer #4: The authors present an integrative approach to combine semantic embedding models and network models of scientific literature. The evolution of scientific literature can then be modeled in terms of a stream of dynamic events over time.

The authors made some good use of some of the recent advances in relevant fields and constructed a meaningful pipeline.

In general, the paper is interesting to read and various technical decisions are explained relatively well.

There are some areas I'd like to draw authors' attention for possible clarification and further improvements.

Relevant works cited by the authors are okay, but there are more closely related works that authors may not be aware of. They should be incorporated to strengthen the work. For example, there is a substantial body of the literature that underlines the immediate context of the work, including graph embedding, theories and mechanisms of how science advances, characterizing clusters with keyphrases, and time slicing the literature to trace the evolution.

The role of the concept of metastability in this paper is not adequately developed. In fact, it is so weak that it can be dropped altogether without losing anything.

There are a few other places where clarifications and additional explanations would be helpful. For example, the switch to interdisciplinarity in line 47 on page 2 is rather unanticipated.

Existing ways to measure interdisciplinarity should be better documented and evaluated. For example, gini and Shannon entropy were mentioned but it is not clear why betweenness centrality is chosen.

Page 3, the beginning of the paragraph mentioned that the dataset contains 19,177 papers published, but at the end of the paragraph, the complete dataset includes 839,096 papers, which can be confusing. How many papers in the dataset and how many citations? Do you mean the 839,096 are references cited by the 19,177 papers in a total of 1.45 million instances? If this is indeed the case, it would be helpful to differentiate the 19,177 papers vs 839,096 papers by calling the latter as references.

DBCV needs to be defined.

The way you are talking about interdisciplinarity can be misleading. It seems more similar to something like inter-cluster connectivity, i.e. more precisely these are cluster-based, whereas it would be questionable to claim these clusters are discipline equivalent. I'd recommend you consider a more precise name for the metric in this context.

The use of the text-based dynamic event taxonomy is interesting.

7. PLOS authors have the option to publish the peer review history of their article (what does this mean?). If published, this will include your full peer review and any attached files.

Reviewer #2: No

Reviewer #3: **Yes: **Adrian Barnett

Reviewer #4: No

---

## [Author Response · Author response to Decision Letter 1]

18 Feb 2023

We thank the reviewers for their thoughtful and helpful comments. We have implemented a number of revisions to the work and the paper based on these suggestions, and the paper is stronger now because of it. Following, please find point-by-point responses to each of the reviewers' comments.

Reviewers' comments

Reviewer's Responses to Questions

Comments to the Author

1. If the authors have adequately addressed your comments raised in a previous round of review and you feel that this manuscript is now acceptable for publication, you may indicate that here to bypass the “Comments to the Author” section, enter your conflict of interest statement in the “Confidential to Editor” section, and submit your "Accept" recommendation.

Reviewer #2: All comments have been addressed

Reviewer #3: (No Response)

Reviewer #4: (No Response)

2. Is the manuscript technically sound, and do the data support the conclusions?

Reviewer #2: Yes

Reviewer #3: Yes

Reviewer #4: Yes

3. Has the statistical analysis been performed appropriately and rigorously? 

Reviewer #2: Yes

Reviewer #3: Yes

Reviewer #4: Yes

4. Have the authors made all data underlying the findings in their manuscript fully available?

Reviewer #2: Yes

Reviewer #3: No

Reviewer #4: No

Response: We have provided the DOIs and titles corresponding to all seed papers analysed in the study at \\url{https://github.com/QS-2/VESPID/blob/main/seed_data/score_papers.csv}. While we are not able to directly share the raw data (titles and abstracts) used for our analysis due to our licensing agreement with Web of Science, this list of DOIs is sufficient to replicate our findings, either directly using Web of Science (for anyone with a license) or indirectly using Semantic Scholar, which is open source. This is described in the Data Sharing section of the manuscript.

5. Is the manuscript presented in an intelligible fashion and written in standard English?

Reviewer #2: Yes

Reviewer #3: Yes

Reviewer #4: Yes

Response to Review Comments to the Author

Reviewer #2: This is my short review of the first revision of the manuscript “The evolution of scientific literature as metastable knowledge states”. In my opinion, the authors addressed the comments I left in my first review. From the technical standpoint, paper looks more explicit and transparent now.

However, I still have some additions considering the manuscript:

1. When the citation network is constructed, the authors use a larger dataset of papers because they aim to capture the information about references. But they only use the reference section of the papers (from what I understand from this sentence: “For citation network analyses, we also included all papers referenced by these papers.”). My concern is that this approach may not be comprehensive because it does not contain the information about the incoming connections (WHO cites a given paper, e.g. “Cited by” button in Google Scholar). I understand that this data may be harder to obtain and process, but that this improvement could be beneficial for citation network analysis.

Response: Thank you for raising this concern. Our reasoning behind not including the incoming citations to a publication is because we are using historical information to forecast the state of a cluster/discipline in the following year. Since it takes time for citations to accumulate, we assume that publications' incoming citations wouldn't be known at the time of forecasting.

2. I would like to get a better understanding of “weak members” of the clusters. Since the authors mention that “HDBSCAN refers to these non-confident assignments as noise”, which is true, they also have expectations about novelty these outliers can bring: “We expect these not to be noise in the traditional sense (e.g., an outlier or data worthy of discarding as it provides no analytical value) but instead to potentially add value as novel research.” I would like the authors to emphasize on this claim and maybe provide some examples of their point of view and why they think it is valid. I personally would like to see examples, where these “weak members” at timestamp t form a stable cluster at timestamp t+1. It is also interesting to investigate their stability: how they behave over time and over different clustering (UMAP + HDBSCAN) runs. These papers, being outliers, can simply be a result of some edge cases for pre-trained embeddings. For example, some research areas could be covered worse than others resulting in underrepresented “spots” in the latent space.

Response: Thank you for this insightful comment.

 1. Our assumptions about novelty stem from the observation that papers identified as weak members are different from the ones that were confidently assigned to a cluster. By novelty we meant that the content of these weak members is different from other work in their field. For example, one weak member is an article titled Must psychologists change the way they analyze their data? (DOI: 10.1037a0024777) published in Journal of Personality and Social Psychology. Another is titled Strengthening the Experimenter’s Toolbox: Statistical Estimation of Internal Validity (10.1111j.15405907.2011.00576.x) published in American Journal of Political Science. These papers are devoted to statistical methods and are published in venues that predominantly publish practical applications. However, we agree that a claim about novelty requires substantiation beyond anecdotal evidence so we have now modified this discussion in the manuscript.

 2. We expect that weak members at time t1 can lead to a stable cluster at t2 (t2 > t1) only if the ensuing research efforts result in a “critical mass”. We plan to explore this in future work and have included this point within the manuscript’s discussion section.

 3. The UMAP + HDBSCAN clustering pipeline is a standard approach in the literature for clustering text, but the question of its stability is important. To address this question, we have done some testing on a subset of the data to assess the stability of our results. Specifically, we ran the UMAP algorithm with five different random seeds on our 2011 data. To compare results between runs, we calculated the jaccard similarity index. We find a mean similarity of 0.78 (pairwise, between runs) with standard deviation 0.042. Results are fully repeatable (jaccard index of 1) when the random seed is kept constant across the runs.

 4. We agree that the dataset used for pre-training the SPECTER model affects the resulting embeddings. We have now added text acknowledging this as a limitation for this work.

3. One of my previous comments was about large scale experimentation and scaling. By saying that, I mean the size of textual embedding space as well. At this moment, textual embeddings (then dimensionality reduction and clustering) are applied to only less than 20,000 papers, which I cannot refer to as a “large scale” experiment (I am expecting at least a million papers there). The citation network scales differently, but it is still a relatively small (and very sparse) graph. It would be interesting if the authors could replicate their results using some faster clustering algorithm (i.e., k-means) and experiment with larger datasets in their future work.

Response: Thank you for these comments.

We agree that the scale of the current dataset is relatively small compared to general machine learning benchmarks in other domains. In ongoing work, we are streamlining the computational pipeline as well as collecting more data with an aim to similarly investigate the full Web of Science corpus.

• We also agree that future work should compare clusters output via our approach to other benchmark clustering algorithms. To our knowledge, there would not be a clear “best” candidate. In addition to requiring an a prior input value for k, k-means clustering assumes the clusters to be spherical which may not be appropriate in the case of high-dimensional knowledge clusters. However, these questions are important to study more formally and we have included a note on this in our discussion section.

Verdict: I think that the authors made some improvements with their first revision, but there are still several things to address before the manuscript can be fully accepted. If they believe that more experimentation is not feasible/realistic/necessary to perform, I would gladly ask them to add some comments on why the particular decisions were made.

Reviewer #3: This paper uses a large data set of papers and citations to examine how scientific ideas evolve over time. The paper was well explained with interesting figures. The overall results made sense and were interesting.

1. There was no mention of the common errors in citations in the limitations. Likely this measurement error would not be a big issue given the sample size, but it should be mentioned. See for example, DOI: 10.1042/CS20201573.

Response: We appreciate the comment we have now added text related to the errors in citation analysis. 

2. The reduction in clusters over time (figure 5) could be because of movements outside the selected fields that are not captured. For example, for the not included field “Computer Science” there could have been a cluster building in the early years that was then joined by researchers from one of the selected fields. I don’t believe that this kind of movement would be captured by this study design. It is counter-intuitive that ideas have decreased over time, and no explanation is given.

Response: We thank the reviewer for raising this idea.

• Yes, we have constrained our analyses to the social sciences and we agree that movements from outside this set of fields could be causing shifts within our data that we may not be able to observe, e.g., increasing integration between computer science and social sciences. We have added this to our discussion of limitations.

• We think that reduction of ideas, at the final years of dataset, may be a misunderstanding. The number of language clusters, which represent ideas, hovers roughly around 50, except for 2015 (see Figure 5). However, the number of network communities do decrease for the last two years in the dataset, which we believe is what the reviewer is referring to. We presented the number of network communities to demonstrate that there is some but not a complete overlap between the network communities and the language clusters. However, the network communities are not otherwise used in our analyses.

3. There are a lot of analytical choices in the paper, hence there is likely to be some model uncertainty (DOI: 10.1111/j.1467-9574.2012.00530.x). Other researchers may have made different and equally plausible decisions that could give different results. I think this should be acknowledged as a limitation.

Response: We have now added the limitations to the discussion section.

4. This is not multinomial logistic regression (line 248) as that is for outcomes with multiple classes. The outcome here is binary, so this is simply logistic regression.

Response: Thank you for pointing this out. We have updated the manuscript.

5. The authors are not compliant with the PLOS ONE data sharing policy: “The PLOS Data policy requires authors to make all data underlying the findings described in their manuscript fully available without restriction, with rare exception”

Response: We have provided the DOIs and titles corresponding to all seed papers analysed in the study at https://github.com/QS-2/VESPID/blob/main/seed_data/score_papers.csv. While we are not able to directly share the raw data (titles and abstracts) used for our analysis due to our licensing agreement with Web of Science, this list of DOIs is sufficient to replicate our findings, either directly using Web of Science (for anyone with a license) or indirectly using Semantic Scholar, which is open source. This is described in the Data Sharing section of the manuscript.

Minor comments

1. Line 36 “state-of-the-art” is hyperbole. Point out the benefits of the methods. 

Response: We have now modified the language around the claim.

2. Line 80, why were these fields chosen? 

Response: We have chosen these disciplines because the dataset was carefully curated through stratified sampling for a different project, Systematizing Confidence in Open Research and Evidence (DARPA SCORE), that aimed to ensure an accurate representation of the disciplines within the social and behavioral sciences.

3. What is embedding? Line 98.

Response: We have now modified the language.

4. SPECTER may “outperform” other methods, but how accurate is it? Line 101

Response: Our assertion relating outperforming comes from the findings provided in the SPECTER article (DOI 10.48550arXiv.2004.07180). However, we acknowledge the need for clearer language and have consequently revised the statement.

5. There are a lot of acronyms in the paper. I am fairly certain that DBCV is not defined. 

Response: We apologize for the lack of definition. We have now added it to the manuscript.

6. The overall proportion of weak members would be interesting (figure 2 legend)

Response: We have now added this information to the legend (approx 11.06%).

7. Figure 2 was interesting, but I didn’t know what the x- and y-axes were. 

Response: We have now added the text in legend. Since the x- and y-axes for this figure were created by projecting the semantic embeddings to two dimensions using UMAP, the axes are no longer interpretable.

8. Could some joins between fields be due to commonly used methods papers?

Response: That is an interesting idea. We agree that the merge event could happen due to shared methods. We hypothesize that the commonality between the fields should reach a certain ”critical mass” for merging to occur and plan to analyze this in a future study.

9. I found it surprising that the death of an area was labelled as “stable”, as that seems like a definite change. This is just a wording issue.

10. Figure 4. The paper for the top-right circle is missing a title. 

Response: Thank you for pointing this we have now added a title for the publication node in the schematic diagram.

11. The splits are based on annual data. That is probably suitable, but should be mentioned as a limitation.

Response: We agree that the choice of one year as a time step fails to capture the evolutionary dynamics with cycles less than a year. We have now added this as a limitation.

12. “features were not significant” (table 1 legend) best to focus on the size of the difference rather than the statistical significance (DOI 10.1080/00031305.2016.1154108)

Response: Thank you for this point and the reference. We assume by ”focusing on the size of the difference” you mean the difference in the effect sizes. If so, we would need to threshold values for selection based on the marginal effects while considering the standard errors simultaneously. We have added this in the manuscript as a limitation.

13. “including only features shown to be statistically significant” there are better model selection methods available than using statistical significance, see for example Statistical Learning with Sparsity: the Lasso and Generalizations by Trevor Hastie, Robert Tibshirani and Martin Wainwright (May 2015)

Response: Our approach to feature selection is commonly used for our choice of the model - specifically, logistic regression (ISBN:9780470582473). We agree and have noted in the manuscript that future work should address the important question of choosing the best statistical model to capture these phenomena.

Reviewer #4: The authors present an integrative approach to combine semantic embedding models and network models of scientific literature. The evolution of scientific literature can then be modeled in terms of a stream of dynamic events over time.

The authors made some good use of some of the recent advances in relevant fields and constructed a meaningful pipeline.

In general, the paper is interesting to read and various technical decisions are explained relatively well.

There are some areas I'd like to draw authors' attention for possible clarification and further improvements.

1. Relevant works cited by the authors are okay, but there are more closely related works that authors may not be aware of. They should be incorporated to strengthen the work. For example, there is a substantial body of the literature that underlines the immediate context of the work, including graph embedding, theories and mechanisms of how science advances, characterizing clusters with keyphrases, and time slicing the literature to trace the evolution.

Response: We have now added literature as suggested.

2. The role of the concept of metastability in this paper is not adequately developed. In fact, it is so weak that it can be dropped altogether without losing anything.

Response: We appreciate this point. We use the term metastability to offer a simple intuition for motivating ideas of this work. That is, the emergence of interdisciplinary work signals that the landscape is shifting within and around a field of study. To our knowledge, there is no better name for this idea. However, if the reviewer would prefer we can remove it.

3. There are a few other places where clarifications and additional explanations would be helpful. For example, the switch to interdisciplinarity in line 47 on page 2 is rather unanticipated.

Existing ways to measure interdisciplinarity should be better documented and evaluated. For example, gini and Shannon entropy were mentioned but it is not clear why betweenness centrality is chosen.

Response: That you for these comments, we have modified the manscript accordingly. Specifically:

 1. We have modified the text there (line 47) to better transition to interdisciplinarity.

 2. While we referenced the Shannon entropy and Gini index, we have developed our own measure of interdisciplinarity based on the semantic information. We took this approach since the interdiciplinary metrics based on Gini index as well as Shannon entropy use predermined subject categories (e.g. Web of Science categories) of the references of the publication rather than their semantic information. Moreover, for the network based interdisciplinarity metric we used the betweenness centrality because it relies on citation networks which cannot be captured by other metrics such as Gini index.

4. Page 3, the beginning of the paragraph mentioned that the dataset contains 19,177 papers published, but at the end of the paragraph, the complete dataset includes 839,096 papers, which can be confusing. How many papers in the dataset and how many citations? Do you mean the 839,096 are references cited by the 19,177 papers in a total of 1.45 million instances? If this is indeed the case, it would be helpful to differentiate the 19,177 papers vs 839,096 papers by calling the latter as references.

Response: Thank you for the comment. We agree that the prior description was unclear. Our study analyzes our core dataset, containing 19,177 publications. We have added the references cited by the 19,177 publications in our dataset to study the citation context for network analysis. We have now added a clear explanation to the manuscript.

5. DBCV needs to be defined.

Response: Thank you. We apologize for neglecting to include this definition. We have now added it to the manuscript.

6. The way you are talking about interdisciplinarity can be misleading. It seems more similar to something like inter-cluster connectivity, i.e. more precisely these are cluster-based, whereas it would be questionable to claim these clusters are discipline equivalent. I'd recommend you consider a more precise name for the metric in this context.

Response: Thank you for the thoughtful point. We acknowledge that the clusters may not represent disciplines. There is no consensus in the research community about the definition and operationalization of interdisciplinarity (see, e.g., https://doi.org/10.1162/qss_a_00011). Nevertheless, it is widely appreciated that interdisciplinarity is concerned with the integration of knowledge. Thus, we expect that our proposed metric can serve as an effective means of measuring interdisciplinarity, without relying on preconceived disciplinary boundaries. Usage of pre-defined subject categories can be a limitation particularly in light of studies indicating that scientific research is becoming more interdisciplinary (https://doi.org/10.1007/s11192-008-2197-2). For instance, a research article about usage of machine learning methods in bridge design can be classified as belonging to the pre-defined category of civil engineering if published in a journal specific to civil engineering. These subtle distinctions, which are frequently overlooked in the context of conventional disciplinary categories, can be captured by our proposed metric. To provide this context to the readers, we have updated the manuscript.

The use of the text-based dynamic event taxonomy is interesting.

7. PLOS authors have the option to publish the peer review history of their article (what does this mean?). If published, this will include your full peer review and any attached files.

Do you want your identity to be public for this peer review? For information about this choice, including consent withdrawal, please see our Privacy Policy.

Reviewer #2: No

Reviewer #3: Yes: Adrian Barnett

Reviewer #4: No

---

## [Decision Letter · Decision Letter 2]

21 Mar 2023

PONE-D-22-12774R2The evolution of scientific literature as metastable knowledge statesPLOS ONE

Dear Dr. Koneru,

Thank you for submitting your manuscript to PLOS ONE. After careful consideration, we feel that it has merit but does not fully meet PLOS ONE’s publication criteria as it currently stands. Therefore, we invite you to submit a revised version of the manuscript that addresses the points raised during the review process.

We look forward to receiving your revised manuscript.

Kind regards,

Ilya Safro, Ph.D.

Academic Editor

PLOS ONE

Journal Requirements:

Reviewers' comments:

Reviewer's Responses to Questions

**Comments to the Author**

1. If the authors have adequately addressed your comments raised in a previous round of review and you feel that this manuscript is now acceptable for publication, you may indicate that here to bypass the “Comments to the Author” section, enter your conflict of interest statement in the “Confidential to Editor” section, and submit your "Accept" recommendation.

Reviewer #2: All comments have been addressed

Reviewer #3: (No Response)

2. Is the manuscript technically sound, and do the data support the conclusions?

Reviewer #2: Yes

Reviewer #3: Yes

3. Has the statistical analysis been performed appropriately and rigorously? 

Reviewer #2: Yes

Reviewer #3: Yes

4. Have the authors made all data underlying the findings in their manuscript fully available?

Reviewer #2: Yes

Reviewer #3: Yes

5. Is the manuscript presented in an intelligible fashion and written in standard English?

Reviewer #2: Yes

Reviewer #3: Yes

6. Review Comments to the Author

Reviewer #2: This is my short review of the second revision of the manuscript “The

evolution of scientific literature as metastable knowledge states”. In my opinion, the

authors adequately addressed the comments I (and other reviewers) left in previous review iterations. Overall, the manuscript looks good to me. Authors explained their experiment design choices, limitations and proposed additional research directions in the “Discussion” section.

Reviewer #3: The authors use a large dataset to estimate how knowledge evolves over time. The paper has a good flow and is well argued. There is a necessarily complex data preparation and analysis given the size of the data and the many model choices. The authors gave thoughtful answers to my previous questions.

Some of the choices are arbitrary, for example matching clusters based on the cosine similarity of 0.95 or higher. Would things have been different if this had been 0.90 or 0.99? It may be useful to do some random checks of the predictions to give confidence that the approach is working as expected. For example, randomly sampling papers and potential clusters, and asking blinded experts to classify the papers and then comparing this with the classifications made by the model.

7. PLOS authors have the option to publish the peer review history of their article (what does this mean?). If published, this will include your full peer review and any attached files.

Reviewer #2: No

Reviewer #3: **Yes: **Adrian Barnett

---

## [Author Response · Author response to Decision Letter 2]

8 May 2023

We thank the reviewers for their thoughtful and helpful comments. We have implemented a number of revisions to the work and the paper based on these suggestions. Following, please find point-by-point responses to each of the reviewers’ comments. 

Reviewer’s Responses to Questions 

Comments to the Author

1. If the authors have adequately addressed your comments raised in a previous round of review and you feel that this manuscript is now acceptable for publication, you may indicate that here to bypass the “Comments to the Author” section, enter your conflict of interest statement in the “Confidential to Editor” section, and submit your ”Accept” recommendation.

Reviewer #2: All comments have been addressed

Reviewer #3: (No Response)

2. Is the manuscript technically sound, and do the data support the conclusions? The manuscript must describe a technically sound piece of scientific research with data that supports the conclusions. Experiments must have been conducted rigorously, with appropriate controls, replication, and sample sizes. The conclusions must be drawn appropriately based on the data presented.

Reviewer #2: Yes

Reviewer #3: Yes

3. Has the statistical analysis been performed appropriately and rigorously?

Reviewer #2: Yes

Reviewer #3: Yes

4. Have the authors made all data underlying the findings in their manuscript fully available? The PLOS Data policy requires authors to make all data underlying the findings described in their manuscript fully available without restriction, with rare exception (please refer to the Data Availability Statement in the manuscript PDF file). The data should be provided as part of the manuscript or its supporting information, or deposited to a public repository. For example, in addition to summary statistics, the data points behind means, medians and variance measures should be available. If there are restrictions on publicly sharing data—e.g. participant privacy or use of data from a third party—those must be specified

Reviewer #2: Yes

Reviewer #3: Yes

5. Is the manuscript presented in an intelligible fashion and written in standard English? PLOS ONE does not copyedit accepted manuscripts, so the language in submitted articles must be clear, correct, and unambiguous. Any typographical or grammatical errors should be corrected at revision, so please note any specific errors here.

Reviewer #2: Yes

Reviewer #3: Yes

6. Review Comments to the Author

Reviewer #2: This is my short review of the second revision of the manuscript “The evolution of scientific literature as metastable knowledge states”. In my opinion, the authors adequately addressed the comments I (and other reviewers) left in previous review iterations. Overall, the manuscript looks good to me. Authors explained their experiment design choices, limitations and proposed additional research directions in the “Discussion” section.

Reviewer #3: The authors use a large dataset to estimate how knowledge evolves over time. The paper has a good flow and is well argued. There is a necessarily complex data preparation and analysis given the size of the data and the many model choices. The authors gave thoughtful answers to my previous questions.

Review: Some of the choices are arbitrary, for example matching clusters based on the cosine similarity of 0.95 or higher. Would things have been different if this had been 0.90 or 0.99?

Response: To better understand the impact of threshold choice for cluster matching, we have carried out a series of experiments with varying thresholds. As one would expect, increasing the threshold (tightening the criteria for a cluster at time t to be considered the same as a cluster at time t + 1) leads to a decrease in the number of identified merges and splits. And consequently, an increase in the number of continuation and death events. Threshold values above 0.97 result in no merge events while lowering the threshold below 0.88 results in no continuation events. Figure 1 gives a summary of impacts of threshold choice on observed events. In this way, we can interpret the cluster similarity threshold as a parameter modulating selection from amongst a hierarchical family of events varying in resolution. 

With respect to prediction task performance, performance is relatively stable for values between 0.88 and 0.91. However, increasing the threshold above 0.91 led to reduction in the performance in our dataset. We hypothesize this may be due to the reduction in number of events identified and thus less training. 

We have added the full results of these additional experiments and discussion of these findings to the paper’s Appendix and linked it in the paper (see Text-based dynamic event modeling).

Review: It may be useful to do some random checks of the predictions to give confidence that the approach is working as expected. For example, randomly sampling papers and potential clusters, and asking blinded experts to classify the papers and then comparing this with the classifications made by the model.

Response: We appreciate the suggestion. During this phase of revision, per the reviewer’s recommendation, we asked blinded experts to classify papers from a randomly selected subset of our clusters and compare the results. Specifically, we asked three faculty members for input - one each from the fields of Marketing, Psychology, and Political Science. We asked them to complete two tasks.

Task 1: We showed each expert titles and abstracts corresponding with a collection of publications in their field (13 marketing papers, 12 each for psychology and political science). We asked them to group those papers into a fixed set of clusters (4 for marketing, 3 for psychology and political science).

Task 2: After they had completed their own clustering, we showed them the clusters our algorithm identified. We asked them to rate the clusters using a Likert scale (1 to 5, where 1 represents extremely bad and 5 represents extremely good).

Results are summarized in Table 1. We use Jaccard similarity to score similarity between the expert-generated clusters and our clusters (Task 1). Despite variation in this outcome across the three fields, experts consistently gave high ratings when evaluating the clusters created by our model (Task 2). This appears to highlight the inherent challenge of coming up with a universal ”ground truth” for defining similar papers or groups of papers in the literature.

Our survey also included an open field for any additional feedback. Following is the feedback we received:

Marketing: One paper (13) is pure theory and more Econ but the others are data driven marketing papers 

Psychology: The B category hangs well together with health and intervention work. It is the A and C groups that are more challenging as they cut across social and cognitive sciences. e.g., paper 6 is about collective cognition which should likely be paired with paper 12 

Political Science: This was perfect – but imo this one was pretty easy, I could do the clustering through the titles alone

Of note, the faculty member from Psychology defined clusters based on normative subfields within the discipline. Conversely, the faculty member from Marketing used differences in the theoretical vs. applied focus of the work as a primary criteria. These anecdotes offer insight into why previous work in science of science has generally relied on subject categories generated by sources like Web of Science as ground truth. However, we argue that existing categories are very high level and do not adequately account for the interdisciplinary nature of much of the literature. We have added to the paper a discussion of the outstanding need for benchmark datasets for taxonomization of interdisciplinary work (see Discussion).

Table 1: Summary of results from a brief survey of three domain experts.

3

Discipline |# clusters |# publications |Task 1 Mean Jaccard similarity |Task 2 Evaluation of algorithm-generated clusters (scale of 1-5)

Marketing |4 |13 |0.46 |4

Psychology |3 |12 |0.7 |4

Political Science |3 |12 |0.88 |5

---

## [Editor Report · Decision Letter 3]

2 Jun 2023

The evolution of scientific literature as metastable knowledge states

PONE-D-22-12774R3

Dear Dr. Koneru,

We’re pleased to inform you that your manuscript has been judged scientifically suitable for publication and will be formally accepted for publication once it meets all outstanding technical requirements.

Kind regards,

Ilya Safro, Ph.D.

Academic Editor

PLOS ONE
---

## [Editor Report · Acceptance letter]

15 Jun 2023

PONE-D-22-12774R3 

The evolution of scientific literature as metastable knowledge states 

Dear Dr. Koneru:

I'm pleased to inform you that your manuscript has been deemed suitable for publication in PLOS ONE. Congratulations! Your manuscript is now with our production department. 

Kind regards, 

on behalf of

Dr. Ilya Safro 

Academic Editor

PLOS ONE